# Validating the bifactor structure of the Ruminative Thought Style Questionnaire—A psychometric study

**Lilla Nóra Kovács**[1,2☯], **Natália Kocsel**[2☯], **Attila Galambos**[1,2], **Anna Magi**[1,2], **Zsolt Demetrovics**[2], **Gyöngyi Kökönyei**[2,3,4]*

**1** Doctoral School of Psychology, ELTE Eötvös Loránd University, Budapest, Hungary, **2** Institute of Psychology, ELTE Eötvös Loránd University, Budapest, Hungary, **3** SE-NAP2 Genetic Brain Imaging Migraine Research Group, Hungarian Academy of Sciences, Semmelweis University, Budapest, Hungary, **4** Faculty of Pharmacy, Department of Pharmacodynamics, Semmelweis University, Budapest, Hungary

☯ These authors contributed equally to this work.
* kokonyei.gyongyi@ppk.elte.hu

**Data Availability Statement:** Data are available at https://osf.io/c6zkd/. DOI: 10.17605/OSF.IO/C6ZKD.

## Abstract

The Ruminative Thought Style Questionnaire (RTSQ) is a self-report measure that aims to capture rumination globally, unbiased by depressive symptoms. We explored its psychometric properties among university students (N = 1123), as the existing models about the factor structure of the RTSQ have been inconclusive. In a second study (N = 320) we tested its convergent validity compared to the Ruminative Response Scale (RRS) and its construct validity compared to the Zung Self-rating Depression Scale (ZSDS). The results of Study 1 suggest that the factor structure of the RTSQ is best described with a 19-item bifactor Exploratory Structural Equation Modelling (ESEM), where most of the variance is explained by the general factor. The model was found to be invariant across genders. The correlations in Study 2 demonstrated that the RTSQ is congruent with the RRS, and that rumination captured by the RTSQ is rather maladaptive, as it was more strongly associated with the brooding subscale of the RRS than with reflective pondering. Significant positive associations were found with depressive symptoms, reaffirming the validity of the RTSQ due to the well-known association between rumination and depressive symptoms. Our results support that RTSQ assesses rumination globally, and it is a valid measure of ruminative thinking style that is rather negatively valenced but does not solely focus on depressive mood and symptoms.

## Introduction

Rumination has become crucial in comprehending negative emotional states and depressive symptoms [1]. A gold standard or consensus about the definition of rumination is lacking. As Smith and Alloy [1] in their review pointed out, there are many different conceptualizations of rumination. These theories differ in several dimensions: a) the degree to which they consider rumination as a stable construct or as a transitive, state-like phenomenon [2,3]; b) whether it is

**Funding:** The first study was supported by the Hungarian National Research, Development and Innovation Office (Grants K111938 and KKP126835). The second study was supported by the Hungarian National Research, Development and Innovation Office (Grant No. FK128614). NK was supported by the ÚNKP-20-4 New National Excellence Program of the Ministry for Innovation and Technology from the source of the National Research, Development and Innovation Fund (Grant No. ÚNKP-20-4-II-ELTE-419). The preparation of this article for GK was supported by the MTA-SE-NAP B Genetic Brain Imaging Migraine Research Group, Hungarian Academy of Sciences, Semmelweis University (Grant No. KTIA_NAP_13-2-2015-0001); Hungarian Brain Research Program (Grant No. 2017-1.2.1-NKP-2017-00002), and by the Higher Education Institutional Excellence Programme of the Ministry for Innovation and Technology in Hungary, within the framework of the Neurology thematic programme of the Semmelweis University (FIKP), and the Research Excellence Programme of the Ministry for Innovation and Technology in Hungary, within the framework of the Biotechnology thematic programme of the Semmelweis University (TKP). The funders had no role in study design, data collection and analysis, decision to publish, or preparation of the manuscript.

**Competing interests:** The authors have declared that no competing interests exist.

the frequency or the actual content of ruminative thought that is more important [4]; c) how rumination relates to other similar or partially overlapping constructs such as negative automatic thoughts, repetitive negative thinking or self-focused attention [3,5]. According to the Response Style Theory (RST) [6], rumination is the tendency to passively and repetitively dwell on one's own depressed mood, concentrating on the possible reasons and consequences of the distress. The RST has gained broad empirical support throughout the past three decades: it has been demonstrated that depressive rumination further increases depressive symptoms [5,7], predicts the commencement [8] and reappearance [9] of depressive episodes, and correlates with their severity [10]. While these empirical findings have often been carried out on community samples, there is a growing body of evidence among clinical populations that confirms that the findings are applicable for patients diagnosed with affective disorders as well [7]. Women are twice as likely to experience depression during their lives than men [11], a gender difference that, according to the RST, might be rooted in females' tendency to react with rumination to stressors [6], while men tend to use other strategies, such as social support or drinking [12]. Ruminative response, accompanied by other psychosocial factors, appears to aggravate depressed mood from early adolescence in case of women [13]. The theory has gained substantial empirical support, as the difference between men and women remained unchanged even when controlling for current depressive symptoms, indicating that elevated ruminative tendencies do not simply occur in response to intensified depressed mood [14].

The RST is certainly the most well-known and most extensively investigated conceptualization of rumination [15], and the Ruminative Response Scale (RRS)—derived from the Response Style Questionnaire (RSQ) [16] that is based on this theory—is the most widely used self-report rumination measure. The RRS has been criticized of being biased by items related to depressive symptoms [1], which led to the removal of such items, thus a shortened version of the scale with two facets was created (brooding and reflective pondering).

Since the RST conceptualized depressive rumination as a possible response to depressed/low mood [6], the RRS items refer to those thoughts and behaviors that make someone focus on their negative emotional state. Obviously, rumination is not restricted to low/depressive mood; other negative emotions or events can induce ruminative thoughts in everyday life or in the laboratory as well. Empirical studies have demonstrated that rumination is linked with other forms of negative affect beyond sadness and depression [17], such as anger [18], shame, guilt [19], or feelings of inadequacy after a social situation [20]. Moreover, although there are fewer studies addressing the relationship between rumination and positive affect, results suggest that positive emotional states may also trigger ruminative responses [21]. These findings support the relevance of defining rumination more broadly, as outlined by Martin & Tesser [22], who proposed that rumination shall be considered as a broad style of thought processing, where the content, valence and even the temporal direction of ruminative thoughts are less important, allowing to extend the domain of rumination-related research.

Building on Martin & Tesser's conceptualization, Brinker & Dozois [7] constructed the Ruminative Thought Style Questionnaire (RTSQ), a 20-item self-report scale that can measure ruminative thoughts without being biased by their valence and temporal orientation. The RTSQ contains items that refer to the present or the future (e.g."When I am anticipating an interaction, I will imagine every possible scenario and conversation."), as well as neutral or positive items (e.g. "When I am looking forward to an exciting event, thoughts of it interfere with what I am working on."). Furthermore, while numerous items of the 22 item RRS appear to measure symptoms of depression [1,2], and both the 22-item and the 10-item RRS instruct participants to evaluate what they think or do when they feel "down, sad or depressed", the authors of the RTSQ aimed to define rumination as a general thinking style, focusing on its intermittent and intrusive nature rather than on the mood or the content of its occurrence.

This goal is reflected in both the phrasing of the items and the more general instructions, where participants are asked to indicate to what extent these items characterize them without specifying the (depressed) mood state. When examining the factor structure of the RTSQ, the authors found the single-factor solution the most adequate. The retained 20 items showed high internal consistency (Cronbach's α = .92 and Cronbach's α = .87). Regarding convergent validity, the RTSQ demonstrated significantly stronger correlation (r = .64) with the Global Rumination Scale [23] than with the 22-item RRS (r = .31), implying that it successfully assesses ruminative tendencies in general, and does not solely focus on the depressive content of ruminative thoughts. Moreover, the authors of the RTSQ conducted a daily diary study on an undergraduate sample where they found that the RTSQ prospectively predicted depressed mood, even after controlling for baseline depressive symptoms, highlighting the clinical significance of rumination among university students, that appears to be well captured by the RTSQ.

Tanner et al. [24] examined the factor structure of the RTSQ on two large adolescent samples (N = 1181 altogether). They removed five items (items 10,15,16,18,19) and suggested a second-order four-factor solution, with subscales named as Problem-focused thoughts (Items 9,11–14), Counterfactual thinking (Items 5–8), Repetitive thoughts (Items 1–4) and Anticipatory thoughts (Items 17, 20) where the items loaded on the four subscales together formulated a general higher-order rumination factor. Thus, the authors concluded that the RTSQ measures rumination as a rather multidimensional, multifaceted construct. In a recent empirical study Bravo et al. [25] found that the Problem-focused Thoughts subscale of the RTSQ mediated the relationship between depressive symptoms and drinking as a means of coping, supporting the scale's relevance among university students.

While Nolen-Hoeksema narrowly defined rumination as a potential response (or response style) to depressed mood, Tanner and colleagues [24] provided a more integrative definition, highlighting the multifaceted nature of rumination. Similarly to Brinker and Dozois [7], Tanner et al. also argue that repetitivity, intrusiveness or uncontrollability are core elements of rumination which might suggest non-productivity, but they also argued that that in some cases, rumination might be useful in identifying strategies and/or resources to cope with future eventualities [24]. Regarding content validity, the authors of the RTSQ focused on conceptualizing rumination as a generic thought pattern, emphasizing its recurrent and intermittent feature rather than its negative valence and past-oriented tendency. Additionally, the four subscales identified by Tanner and colleagues may reflect those core aspects of rumination that the RTSQ can capture. Past psychometric studies [26,27] consistently found the Repetitive thought subscale (items 1–4), indicating that the RTSQ reflects the repetitive nature of ruminative thinking well. However, other important aspects of rumination, such as automaticity, involuntariness, and goal insensitivity [15] may be reflected less by the items of the RTSQ.

In the past decade, most studies that evaluated the factor structure of the RTSQ either tested the unifactorial model suggested by Brinker and Dozois [e.g. 28,29], the four-factor solution described by Tanner et al. [26,30,31] or examined both [27,32,33]. Studies comparing the single-factor and the second-order four-factor models unequivocally found better model fits for the latter. Mihić et al. [27] however suggested a third alternative, a bifactor model as the best solution, with the possibility to reconcile the unifactorial and the four-factor solutions. Mihić et al. [27] found that once the general factor was controlled for, the four subscales did not contribute to the explained variance of the RTSQ significantly, thus the applicability of the subscales was not fully supported according to their results. A summary of previous studies assessing the factor structure of the RTSQ is demonstrated in Table 1.

**Table 1. Empirical studies assessing the factor structure of the RTSQ in different cultural and linguistic settings.**

| Author/year | Language of RTSQ | Sample(s) N, $M_{age}$ (SD) | Method | Tested/preferred model (N of items), fit indices |
|---|---|---|---|---|
| Brinker & Dozois, 2009 [7] | English | 309 university students, $M_{age}$ = 18.96 (3.72) | PCA | one factor model (20) fit indices: N/A |
| Tanner et al., 2013 [24] | English | 2362 adolescents, $M_{age}$ = 13.95 (0.99) | EFA, PCA, CFA | higher order four factor model (15) $\chi^2$ = 666.49, CFI = 0.95, NFI = 0.94, RMSEA = 0.08, SRMR = 0.07 |
| Karatepe et al., 2013 [28] | Turkish | 262 university students, age not reported | PCA | one factor model (20) fit indices: N/A |
| Claycomb et al., 2015 [26] | English | 304 trauma-exposed primary care patients, $M_{age}$ = 42.56 (11.66) | CFA | four factor model (15) $\chi2$ = 342.51, CFI = 0.97, TLI = 0.97, RMSEA = 0.08 |
| Helmig et al., 2016 [33] | German | 203 nonclinical individuals, $M_{age}$ = 40.6 (12.8); 201 clinical individuals, $M_{age}$ = 36.1 (12.8) | CFA | higher order four factor model (15) nonclinical sample: $\chi2/df$ = 2.17, CFI = 0.98, TLI = 0.98, RMSEA = 0.08; clinical sample: $\chi2/df$ = 1.40, CFI = 0.99, TLI = 0.99, RMSEA = 0.05 |
| Walsh et al., 2017 [29] | English | Australian Sample: 369 university students, $M_{age}$ = 21 (SD not reported); Chinese (English-Chinese bilingual) Sample: 123 university students, $M_{age}$ = 20 (SD not reported) | CFA | one factor model (20) Australian Sample: $\chi2$ = 767.60, CFI = 0.980, TLI = 0.976, RMSEA = 0.102; Chinese Sample: $\chi2$ = 201.20, CFI = 0.988, TLI = 0.986, RMSEA = 0.085 |
| Bravo et al., 2018 [32] | English, Spanish (Spain), Spanish (Argentina) | U.S sample: 924 university students, $M_{age}$ = 21.98 (6.33) Argentinean sample: 403 university students, $M_{age}$ = 22.55 (4.17) Spanish sample: 305 university students, $M_{age}$ = 21.03 (4.08) | CFA | four-factor model (15) U.S sample: $\chi2$ = 308.30, CFI = 0.968, TLI = 0.960, RMSEA = 0.054, SRMR = 0.044; Argentinean sample: $\chi2$ = 271.65, CFI = 0.921, TLI = 0.901, RMSEA = 0.074, SRMR = 0.061; Spanish sample: $\chi2$ = 201.49, CFI = 0.936, TLI = 0.921, RMSEA = 0.068, SRMR = 0.054; |
| Dzhambov et al., 2019 [30] | Bulgarian | 529 university students, $M_{age}$ = 21(2)* | CFA | four factor model (15) $\chi2$ = 253.897, CFI = 0.953, RMSEA = 0.064, SRMR = 0.044. |
| Mihić et al., 2019 [27] | Serbian | heterogeneous adult sample, $M_{age}$ = 26.5(6.44) | CFA | bifactor model (19) $\chi2$ = 633.49, CFI = 0.95, TLI = 0.94, RMSEA = 0.06–0.07, SRMR = 0.04 |
| Tonta et al., 2020 [31] | English | 735 university students, $M_{age}$ = 21.69 (6.12) | CFA | four factor model (15) $\chi2$ = 304.32, RMSEA = 0.060, CFI = 0.963, TLI = 0.953, SRMR = 0.047 |

RTSQ, Ruminative Thought Style Questionnaire; EFA, Exploratory Factor Analysis; PCA, Principal Component Analysis; CFA, Confirmatory Factor Analysis; $\chi^2$, chi-square test statistic; df, degree of freedom; CFI, Comparative Fit Index; TLI, Tucker-Lewis Index; RMSEA, Root Mean Squared Error of Approximation; SRMR, Standardized Root Mean Square Residual.

* Median (interquartile range).

Although results about the cross-cultural validity of the RTSQ are scarce, Bravo et al. and Walsh et al. found that the RTSQ demonstrated measurement invariance across U.S, Spanish and Argentinian samples [32], and across Chinese and Australian samples [29].

Among the factor extraction methods described above, only the bifactor solution is capable of separating how much of the item response variance derives from a single latent variable, and how much is attributable to its subgroups, which is a crucial aspect when improving a scale that may contribute to better interpret the trait itself.

In summary, the psychometric evaluation of the RTSQ thus far has yielded inconclusive results, and the factor structure of the Hungarian RTSQ [34] has not been investigated. Our primary goal was to see whether a strong common trait or factor–rumination—existed behind the different items or factors to see whether the sum score of the RTSQ could be reliably used in future studies. On the other hand, there is substantial heterogeneity in past studies not only in the language of the RTSQ, but also the research methodologies applied. Therefore, another aim of our research was to investigate the factor structure and psychometric properties of the Hungarian RTSQ by testing the models presented above on two demographically more

homogenous adult samples in two different studies. Furthermore, due to the ambiguity of these models, we also aimed to examine the factor structure of the RTSQ with Exploratory Structural Equation Modelling (ESEM), a method that consists of both confirmatory and exploratory features [35]. CFA requires item cross-loadings to be fixed at zero, however, for many measurement models this restriction may be impractical and often contradicts the background theory of the measure [36]. One clear advantage of CFA is the capability to build concise models and it is considered the go-to approach when a solid measurement model is available [37]. ESEM, on the other hand allows for items to load on multiple factors, which may be a more accurate representation of reality when subscales are not entirely independent [38]. Also, when there is a lack of consensus regarding the measurement model, or its structure is more complex (and would be oversimplified by the CFA approach), the use of ESEM is recommended [36]. In case of the RTSQ, theory posits that there is a latent overarching construct, rumination [7], and therefore the assumption that it comprises four independent subscales is highly improbable and such measurement specification may lead to error. Furthermore, given the inconclusive results in the literature, a solid measurement model of the RTSQ is unavailable. Therefore, we also wished to test its factor structure with ESEM.

In Study 1 our goals were to 1) evaluate the degree of fit of the four previously mentioned measurement models of RTSQ; 2) test the best fitting model with ESEM; 3) test the gender invariance of the best fitting model [39]; 4) investigate the psychometric properties of RTSQ and 5) test its construct validity with the Center for Epidemiologic Studies Depression Scale (CES-D) [40] and the Brief Symptom Inventory (BSI) [41,42]. Based on previous results [24,26,43] we expected that the RTSQ would have a significant positive relationship with the CES-D and the BSI scales.

In Study 2 we aimed to test the construct validity of the RTSQ using the short form of the Ruminative Response Scale (RRS) [2], which measures two different facets of rumination: brooding and reflective pondering. Considering previous theoretical and empirical work [27] we hypothesized positive associations between the RTSQ and the reflective pondering and brooding factors of the RRS.

## Study 1

### Methods of Study1

**Sample and procedure.** Two independent researchers translated the RTSQ from English to Hungarian. Differences were resolved by discussion and consensus with the help of a third native Hungarian-speaking researcher who used to live in an English-speaking country for years. Then a fourth researcher backtranslated the Hungarian version to English. A native English-speaking psychologist reviewed the two versions and found that the backtranslation adequately reflected the meaning of the original items.

Data collection was carried out within the framework of a larger research project examining the psychological and genetic factors of addictive behaviors [44]. Ethical consent was obtained from the Scientific and Research Ethics Committee of the Medical Research Council (ETT TUKEB) for the whole research project including this study. Approval number: 20707-0/2010-1018EKU (840/PI/010.) Written informed consent of participants was obtained. Students were recruited from several university dormitories, who participated in the study on a voluntary basis. Potential participants were contacted in person in their dormitories by research assistants in a systematic manner, where they could fill out the self-report measures on paper in their room at their own pace after providing written informed consent. Inclusion criteria were age of 18 years or older and active student status at the university, no further restrictions applied. In all institutions, refusal to participate in the whole study was approximately 5%.

Altogether 1139 university students agreed to participate, however, 16 of them did not fill out the relevant measures, thus they could not be included in the analysis. Therefore, the overall sample of the current study comprised of 1123 university students, with a fairly balanced gender ratio (percentage of female participants = 55%; N = 618), where the minimum age was 18, the maximum 37 years (M = 21.96; SD = 1.96).

**Measures.** *Ruminative Thought Style Questionnaire (RTSQ) [7].* RTSQ is a 20-item self-report scale that is aiming to measure rumination regardless of the valence, temporal orientation, or content of such thoughts. Participants have to respond on a 7-point Likert-scale (*1 = not at all descriptive of me; 7 = describes me very well*) to items such as *"When I am expecting to meet someone, I will imagine every possible scenario and conversation"*. The RTSQ total score has shown excellent internal consistency (Cronbach α = .89 –.92) and high test-retest reliability after two weeks (r = .80, p < .01) [7], as well as its subscales suggested by Tanner et al. [24] (Cronbach α = .71- .89). The Hungarian RTSQ also demonstrated high internal consistency in two independent studies (Cronbach α = .88; Cronbach α = .91) [45,46].

*The Center for Epidemiologic Studies Depression Scale (CES-D) [40].* The CES-D has been designed for measuring depressive mood in the general population [40]. The original 20-item instrument was shortened to eight negative affect items (e.g. *"I felt lonely"; "I felt fearful"*) and two positive affect items (e.g. *"I felt hopeful about the future"; "I was happy"*). Participants are asked to evaluate on a four-point Likert scale from *0 = never* to *3 = always* how often they felt this way during the last seven days. The two positive affect items were reversed when calculating the sum score of the scale. The test's Hungarian adaptation demonstrated good internal consistency in a previous study (Cronbach α = .82) [47], as well as in this sample (Cronbach α = .77).

*The Brief Symptom Inventory (BSI).* [41,42] primarily aims to measure psychological symptoms of clinical patients. The BSI is the shortened form of the Derogatis Symptom Checklist (SCL-90) [48] that consists of nine subscales, measuring symptom domains on a five-point Likert scale ranging from *0 = not at all to 4 = extremely*. The mean score of the 53 items is referred to as the General Symptom Index (GSI). In a previous study, the Hungarian adaptation of the BSI demonstrated a bifactor solution with a solid global factor comprised of all items, where the subscales contributed little to the explained variance [47]. Hence, we only included the GSI in our analyses, which demonstrated excellent internal consistency in the current sample (Cronbach α = .95).

**Data analysis strategy.** Data was analyzed using SPSS 25.0 (IBM SPSS, IBM Corp., Armonk, NY) and Mplus 7.4 software packages [49]. Firstly, structural equation modeling (SEM) was performed to estimate the degree of fit of three prior measurement models. The maximum likelihood robust (MLR) parameter estimates were used during the analyses with standard errors and chi-square test statistics that were robust to non-normality and non-independence of observations [50]. Multiple fit indices were considered to evaluate model fit. The index of Root Mean Squared Error of Approximation (RMSEA) below .05 indicates optimal fit, while a value above .10 indicates poor fit. The non-significant value indicates acceptable model fit [51]. Acceptable model fit also requires the Comparative Fit Index (CFI) and the Tucker-Lewis Index (TLI) to be around or higher than .90-.95 [51]. The Standardized Root Mean Square Residual (SRMR) value was also used as an index to assess the fitness of the model, which indicates a good fit below .08 [52]. The tested non-nested models were compared with Aikaike Information Criteria (AIC), where the model with the lowest AIC value was considered as the best fitting model to the data.

In the next stage of analysis, we tested a bifactor ESEM on the bifactor model proposed by Mihić et al [27]. In the bifactor ESEM (Model 4), items loaded on their main factors, but cross-loadings were allowed (targeted, but not forced to be zero). After a thorough inspection of the

items we did not include correlated uniquenesses (i.e. covariances between the error terms of items) to our model. The model fit was evaluated according to the above described criteria. In addition to considering fit indices of the models, the internal consistency of the RTSQ was analysed. Besides Cronbach's alpha, we calculated the omega total coefficient ($\omega$) to examine the proportion of variance in the (unit-weighted) RTSQ total score, attributable to all sources of common variance [53,54]. Based on previous studies [37], the coefficient was calculated as follows: sum of factor loadings$^2$/sum of factor loadings$^2$ + residual variance of items. Furthermore, we estimated the omega hierarchical coefficients ($\omega_h$), which indicates that proportion of the systematic variance in the test's total scores that may be due to between-subject dissimilarities on the general factor, by demonstrating the ratio of the general factor's variance in contrast to the total variance of the measure [55]. According to Reise and colleagues [56], an omega value of .75 or higher would be preferred.

In the next stage of data analysis, we tested the gender invariance of the best fitting model using a multigroup approach in Mplus 7.4. In the configural invariance model the same factor structure and same associations between items and factors were assessed among males and females, without equality constraints. In the metric invariance model, all factor loadings were constrained to be invariant, while in the strong or scalar invariance model both the factor loadings and items' intercepts were set to be equal across gender groups. In a subsequent model, we tested the strict measurement invariance as well, where all factor loadings, intercepts, and items' uniquenesses were constrained to be invariant across males and females. In addition, two further models were tested in which invariance constraints were specified at the level of the factor variances and covariances, and latent means, following the suggestions of Morin et al. [57]. The tested non-nested models were compared with Aikaike Information Criteria (AIC) and Bayesian Information Criteria (BIC). In the past, the model with the lowest AIC or BIC value were considered as the best fitting model to the data, but subsequent studies pointed out that information criteria should be considered as a rough guideline that should be used in combination with parameter estimates and theoretical adequacy, especially outside of the CFA framework, such as ESEM [58,59]. According to previous recommendations, the assumed invariance was accepted if the change in the value of CFI and RMSEA was below or equal to .010 and .015, respectively [60].

Finally, correlation analyses were conducted to test the construct validity of the RTSQ.

## Results of Study 1

**Comparing measurement models.** Four measurement models were compared during the analysis, including 1) the originally proposed one factor model by Brinker and Dozois [7]) (Model 1); 2) the second-order four-factor solution found by Tanner and colleagues [24] (Model 2); 3) and the bifactor model of Mihić and colleagues [27] (Model 3). In the bifactor model of Mihić and colleagues [27] almost every item (except Item 16) loaded to the general rumination factor, but several items were left out of group factors due to low factor loadings (i.e.: items 5,10,14,15,18). As we have outlined in the introduction, we tested a 4) bifactor ESEM as well (Model 4). Thanks to this approach we were able to combine the advantages of the explanatory and confirmatory methods, and we could build a theoretically more suitable model (i.e. in contrast to CFA, in ESEM cross-loadings between the specific factors were targeted but not forced to be 0) [37,57]. Maintaining the factor structure proposed by Mihić and colleagues [27], we formulated one general factor and four specific factors (*Problem-focused thoughts*: Items 9,11,12,13*; Counterfactual thinking*: Items 6–8*; Repetitive thoughts*: Items 1–4*; Anticipatory thoughts*: Items 17,19,20). After a thorough content check, we also decided to leave out Item 16 *("I like to sit and think about pleasant events from the past.")*, which is in line with previous recommendations [24,27].

**Table 2. Factor analyses of four measurement models of the Ruminative Thought Style Questionnaire.**

|  | AIC/BIC | $\chi^2$ | df | CFI | TLI | RMSEA | 90% CI | SRMR |
|---|---|---|---|---|---|---|---|---|
| **Model 1** | 80702.246/81003.672 | 2579.555 | 170 | .699 | .663 | .112 | .11-.12 | .079 |
| **Model 2** | 58942.291/59188.455 | 576.214 | 86 | .916 | .897 | .071 | .07–08 | .059 |
| **Model 3** | 74569.131/74925.818 | 865.870 | 138 | .906 | .883 | .069 | .06-.07 | .047 |
| **Model 4** | 74008.382/74626.304 | 318.861 | 86 | .970 | .940 | .049 | .04-.06 | .020 |

Model 1 = One factor CFA; Model 2 = Second-order four factor CFA; Model 3 = bifactor CFA; Model 4 = bifactor ESEM.

AIC, Akaike Information Criteria; BIC, Bayesian Information Criterion, $\chi^2$, chi-square test statistic; df, degree of freedom; CFI, Comparative Fit Index; TLI, Tucker-Lewis Index; RMSEA, Root Mean Squared Error of Approximation; CI, confidence interval; SRMR, Standardized Root Mean Square Residual.

Table 2 shows the fit indices for each model. Model 1 did not fit the data, while both Model 2 and Model 3 indicated unsatisfactory fit. The only acceptable model was Model 4, implying that the variance was best explained by a bifactor ESEM structure, where 14 out of 19 items loaded on the subfactors besides the general factor. Standardized factor loadings of Model 4 are presented in Table 3.

**Internal consistency of the best fitting model (bifactor ESEM).** The Cronbach αs of the total score of the RTSQ and its subscales demonstrated good internal consistency, in line with previous findings [24,27]. In order to eliminate the errors in the estimation of internal consistency, the omega total and omega hierarchical coefficients were calculated (for details see Table 4).

Given that the omega total of the RTSQ was .939 and the omega hierarchical coefficient for the whole scale was .851 we could assume that only 15% of the total score variance was

**Table 3. Standardized factor loadings of the bifactor ESEM of the RTSQ.**

| Items | Bifactor | RT | CT | PfT | AT |
|---|---|---|---|---|---|
| 1. I find that my mind goes over things again and again | **.50** | **.61** | -.05 | -.04 | -.07 |
| 2. When I have a problem, it will gnaw on my mind for a long time | **.55** | **.61** | -.01 | .07 | -.03 |
| 3. I find that some thoughts come to my mind over and over throughout the day | **.56** | **.63** | -.05 | -.02 | -.05 |
| 4. I can't stop thinking about some things | **.52** | **.38** | .09 | .02 | .09 |
| 5. When I am expecting to meet someone, I will imagine every possible scenario and conversation | **.55** | .08 | .23 | -.09 | .01 |
| 6. I tend to replay past events as I would have liked them to happen | **.49** | -.05 | **.58** | .03 | -.01 |
| 7. I find myself daydreaming about things I wish I had done | **.53** | .01 | **.57** | .05 | -.07 |
| 8. When I feel I have had a bad interaction with someone, I tend to imagine various scenarios where I would have acted differently | **.60** | .00 | **.48** | -.06 | -.03 |
| 9. When trying to solve a complicated problem, I find that I just keep coming back to the beginning without ever finding a solution | **.56** | .00 | .16 | **.23** | -.04 |
| 10. If there is an important event coming up, I think about it so much that I work myself up | **.53** | .07 | .01 | .10 | .43 |
| 11. I have never been able to distract myself from unwanted thoughts | **.56** | .14 | .03 | **.37** | .14 |
| 12. Even if I think about a problem for hours, I still have a hard time coming to a clear understanding | **.49** | -.05 | .03 | **.71** | -.03 |
| 13. It is very difficult for me to come to a clear conclusion about some problems, no matter how much I think about it | **.55** | -.01 | -.09 | **.57** | -.10 |
| 14. Sometimes I realise I have been sitting and thinking about something for hours | **.64** | -.06 | -.06 | .11 | -.17 |
| 15. When I am trying to work out a problem, it is like I have a long debate in my mind where I keep going over different points | **.72** | -.06 | -.22 | -.09 | -.18 |
| 16. When I am looking forward to an exciting event, thoughts of it interfere with what I am working on | **.58** | -.09 | -.03 | .07 | **.41** |
| 17. Sometimes even during a conversation, I find unrelated thoughts popping into my head. | **.56** | -.04 | .04 | .02 | .19 |
| 18. When I have an important conversation coming up, I tend to go over it in my mind again and again | **.65** | .04 | .06 | -.16 | **.28** |
| 19. If I have an important event coming up, I can't stop thinking about it | **.55** | .01 | -.07 | .03 | **.66** |

RTSQ, Ruminative Thought Style Questionnaire; RT, Repetitive thoughts factor of the Ruminative Thought Style Questionnaire; CT, Counterfactual thinking factor of the Ruminative Thought Style Questionnaire; PfT, Problem-focused thoughts factor of the Ruminative Thought Style Questionnaire; AT, Anticipatory thoughts factor of the Ruminative Thought Style Questionnaire.

**Table 4. Alpha and omega internal consistency for the bifactor ESEM of the RTSQ (Model 4).**

| Model 4 | Omega total (ω) | Omega hierarchical (ω$_h$) | Cronbach α |
|---|---|---|---|
| General bifactor | .939 | .851 | .910 |
| RT | .856 | .430 | .843 |
| CT | .806 | .384 | .800 |
| PfT | .826 | .356 | .793 |
| AT | .802 | .231 | .765 |

RTSQ, Ruminative Thought Style Questionnaire; RT, Repetitive thoughts factor of the Ruminative Thought Style Questionnaire; CT, Counterfactual thinking factor of the Ruminative Thought Style Questionnaire; PfT, Problem-focused thoughts factor of the Ruminative Thought Style Questionnaire; AT, Anticipatory thoughts factor of the Ruminative Thought Style Questionnaire.

attributable to the group factors. The omega hierarchical values of the subscales were low compared to omega total values, indicating that the majority of the subscale score variances could be attributed to the general factor and not to the group factors [54].

**Measurement invariance across gender.** The configural model showed a satisfactory fit to the data (see Table 5). Our findings also supported the metric, scalar and strict level gender invariance of bifactor ESEM model as adding constraints to the factor loadings or intercepts did not result in a significant decrease of model fit (according to the recommended cutoff scores of ΔCFI < .010; ΔRMSEA < .015) [60–62]. The invariance model of latent variance-covariance was also supported, but the invariance of latent means was not supported, as the changes of fit indices exceeded the cutoff scores (ΔCFI = -.014). These results indicate that when latent means are constrained to zero in the reference group (males) and are freely estimated in the other group (females), latent means of the female group are significantly higher on the general bifactor (M = .308, p < .001), the problem-focused thoughts (M = .160, p < .05), repetitive thoughts (M = .433, p < .001) and anticipatory thoughts factors (M = .520, p < .001) compared to males.

**Descriptive statistics and construct validity.** Means, standard deviations and effect sizes by gender are shown in Table 6. Significant gender differences were found between the variables, but the Cohen's d values indicated small or medium effects.

In order to test the construct validity of the RTSQ, correlations analysis was conducted (see Table 7 for details). In line with our expectations, the RTSQ showed significant positive correlation both with the CES-D and the BSI scores.

To further investigate the construct validity of RTSQ, we estimated a model with covariates to explore the total score and the subscales' relationship with depression across gender. The

**Table 5. Testing measurement invariance of the RTSQ across genders.**

| Model | χ2(df) | AIC/BIC | RMSEA | RMSEA 90%CI | TLI | CFI | Model comparison | ΔRMSEA | ΔCFI |
|---|---|---|---|---|---|---|---|---|---|
| A.) Configural invariance | 430.787(172)* | 73913.023/75148.868 | .052 | [.046-.058] | .932 | .966 | - | - | - |
| B.) Metric/weak invariance | 526.271(242)* | 73891.454/74775.636 | .046 | [.040-.051] | .947 | .962 | B-A | -.006 | -.004 |
| C.) Scalar/strong invariance | 555.960(256)* | 73893.351/74707.200 | .046 | [.040-.051] | .947 | .960 | C-B | < .001 | -.002 |
| D.) Strict invariance | 555.960(256)* | 73893.351/74707.200 | .046 | [.040-.051] | .947 | .960 | D-C | < .001 | < .001 |
| E.) Var-covariance invariance | 618.019(290)* | 73898.995/74542.036 | .045 | [.040-.050] | .949 | .957 | E-D | -.001 | -.003 |
| F.) Invariance of latent means | 724.556(295)* | 74008.382/74626.304 | .051 | [.046-.056] | .934 | .943 | F-E | .006 | -.014 |

RTSQ, Ruminative Thought Style Questionnaire; χ$^2$, chi-square test statistics; df, degree of freedom; RMSEA, Root Mean Squared Error of Approximation; CFI, Comparative Fit Index; TLI, Tucker-Lewis Index, CI, confidence interval.

*p < .05.

**Table 6. Means and standard deviations of the variables, along with t-statistics and effect sizes by gender.**

| Variables (α) | Total sample, M (SD) | Males, M (SD) | Females, M (SD) | t (p) | | Effect size Cohen's d |
|---|---|---|---|---|---|---|
| RTSQ total (α = .91) | 78.18(20.07) | 73.64(19.27) | 81.86(19.97) | 6.89 (< .001) | 0.42 | |
| PfT (α = .79) | 12.43(5.13) | 11.60(4.72) | 13.11(5.35) | 4.99 (< .001) | 0.30 | |
| CT (α = .80) | 13.11(4.66) | 12.74(4.54) | 13.41(4.74) | 2.41(.02) | 0.14 | |
| RT (α = .84) | 19.99(5.16) | 18.72(5.30) | 21.03(4.80) | 7.63 (< .001) | 0.46 | |
| AT (α = .77) | 12.69(4.17) | 11.59(4.03) | 13.59(4.06) | 8.21 (< .001) | 0.49 | |
| BSI_GSI (α = .95) | 1.68(.50) | 1.60(.45) | 1.74(.53) | 4.99 (< .001) | 0.28 | |
| CES-D (α = .77) | 9.72(4.83) | 9.02(4.59) | 10.29(4.95) | 4.37 (< .001) | 0.27 | |

Total Sample: N = 1123; Males: N = 505 (45%); Females: N = 618 (55%). RTSQ, Ruminative Thought Style Questionnaire; RTSQ subscales: RT, Repetitive thoughts; CT, Counterfactual thinking; PfT, Problem-focused thoughts; AT, Anticipatory thoughts; BSI_GSI, Brief Symptom Inventory General Symptom Index; CES-D, The Center for Epidemiologic Studies Depression Scale; M, mean; SD, standard deviation.

standardized regression weights for the total sample and by gender can be found in S1 Table of the supporting material.

## Discussion of Study 1

In Study 1, we examined four competing models of the RTSQ factor structure based on previous recommendations in the literature on a large sample of university students. Considering the guidelines of Hu & Bentler [63], Model 3 did not demonstrate an adequate fit due to their low CFI and TLI values, thus we could not accept it as our best model. However, when subscales do not represent distinct entities, forcing items to load on one single factor will not represent the construct accurately [37]. RTSQ was aimed to measure rumination globally [7]– assuming that its subscales are not interrelated seems arbitrary and contradicts its theoretical background. ESEM allows for item cross-loadings, thus it is preferred in case of complex scales that lack consensus about their factor structure [35], such as the RTSQ. Additionally, although a bifactor model may not be appropriate for all measures, especially those with homogenous item content, it is considered the best model for those instruments where we *theoretically* expect a strong common trait behind the responses, but also a multidimensionality caused by well-defined clusters [56]. Therefore, we proposed a fourth model, a bifactor ESEM solution containing 19 items on the general factor, and 14 items on the subscales that demonstrated the best model fit. This finding appears to reconcile the original unidimensional suggestion of the authors of the RTSQ [7] with the multifactorial proposition of Tanner et al. [24], in line with

**Table 7. Correlations between RTSQ, BSI and CES-D scores (Study 1).**

| | total sample | | female | | male | |
|---|---|---|---|---|---|---|
| | BSI_GSI | CES-D | BSI_GSI | CES-D | BSI_GSI | CES-D |
| RTSQ total | .53 | .46 | .53 | .44 | .50 | .45 |
| PfT | .50 | .45 | .52 | .44 | .51 | .44 |
| CT | .36 | .33 | .39 | .34 | .30 | .31 |
| RT | .41 | .37 | .46 | .38 | .44 | .38 |
| AT | .35 | .29 | .36 | .27 | .35 | .31 |
| CES-D | .71 | 1.00 | .71 | 1.00 | .69 | 1.00 |

Total Sample: N = 1123; Males: N = 505 (45%); Females: N = 618 (55%). RTSQ total, Ruminative Thought Style Questionnaire total score; RTSQ subscales: RT, Repetitive thoughts; CT, Counterfactual thinking; PfT, Problem-focused thoughts; AT, Anticipatory thoughts; BSI_GSI, Brief Symptom Inventory General Symptom Index; CES-D, The Center for Epidemiologic Studies Depression Scale; M, mean; SD, standard deviation. All correlations are significant at p < .001 level.

the findings of Mihić et al. [27]. Allowing for cross-loadings substantially improved the factor scores of the model too, supporting that this approach represents the structure of the RTSQ better. Moreover, the exploratory feature of the ESEM may unravel where ambiguous items belong to. For instance, by replicating the model proposed by Mihic and colleagues, we only allowed item 10 to be loaded on the general factor, however, the ESEM approach revealed that it may belong to the AT subscale, that is in line with its content ("If there is an important event coming up, I think about it so much that I work myself up.").

Taken together, these results indicate that the RTSQ factor structure can be best described as bifactorial, where the global factor is accountable for most of the explained variance, and the subscales' applicability is limited. However, since the subscales also contributed to the explained variance, and the bifactorial ESEM showed the most adequate model fit, it is unequivocally preferred over the single-factor solution.

We sought to test for the gender invariance of the RTSQ, i.e. whether systematic differences can be found in the way males and females reply to the items. Since most studies applying self-report rumination measures report significant gender differences, presenting that women tend to ruminate more than men [14,64], it is important to examine whether this is attributable to the lack of gender invariance of the measure. Based on the chi-squared difference statistics, the invariance was only supported for the configural model and not for more constrained models. However, given that the chi-square difference test is often criticized because of its sensitivity to the sample size and to normal distribution [51], additional analyses of other indices is worthwhile [62]. Cheung and Rensvold [65] recommended that CFI or RMSEA delta values be investigated before conclusions are made about the lack of invariance. Decreases of .01 or more in CFI across the models provide more certainty that the hypothesis should be rejected [60]. CFI and RMSEA delta values in our sample suggested configural, metric, scalar, strict and var-covariance invariance of our proposed bifactor ESEM, indicating that the RTSQ is a reliable measure across gender. The latent mean score of women was higher, suggesting that women tend to ruminate more than men, and this difference is not attributable to measurement bias [37].

Rumination has been extensively described as a risk factor to the onset, maintenance and relapse of depression [e.g. 5,66]. Recent studies however suggest that rumination is a transdiagnostic risk factor to psychopathology in general, rather than being specific to depressive disorders [67,68], pointing out on one hand that measures not restricted to depressive rumination are required and on the other hand that the outcome of rumination can be diverse. Thus, we wished to test the construct validity of the RTSQ with the help of the CES-D depression scale, and the BSI, that measures psychological symptom patterns in general. In line with our expectations, we found moderate significant correlations between the RTSQ and both clinical scales. Furthermore, the regression model (S1 Table) showed that depressive symptoms were significantly associated with the RTSQ total score and subscales, except for AT, which is plausible given the more positive and future-oriented content of its items. The strongest predictor of depressive symptoms was the RTSQ total score, indicating the relevance of the total score in clinical settings.

## Study 2

### Methods of Study 2

**Sample and procedure.**    In Study 2, our primary goal was to test the construct validity of the RTSQ in order to support the findings of Study 1. Moreover, as that the factor structure of the Hungarian RTSQ had not been examined elsewhere, we considered it important to reexamine it on an independent–albeit relatively small—sample. Undergraduate psychology students were recruited in exchange for partial class credit. Eligibility criteria included being 18 years old or older with no previous history of mental or neurological illness. The students

completed self-report questionnaires online in the computer lab within a bigger study framework for 45 minutes. The study was approved by the Institutional Review Board of the Faculty of Education and Psychology, Eotvos Lorand University (approval number: 2018/396), and data collection was carried out in accordance with the Declaration of Helsinki. Participation in the study was voluntary and anonymous, and written informed consent was obtained. A total of 320 participants (268 females; mean age = 23.28, SD = 2.93 years) could be included for analysis. In Study 2 instead of the CES-D, we applied the Zung Self-Rating Depression Scale, another widely used and reliable measure of depression. Empirical findings support that the two scales are interchangeable [69,70]. Appling another measure of depression enabled us to see whether the findings of Study 1 regarding the associations of RTSQ and depression would be generalizable to another depression scale, reducing the probability that any association between the two constructs is due to item-level biases.

**Measures.** *Ruminative Response Scale (RRS) [2].* The RRS contains 10 items rated on a four-point Likert scale from *1 = never to 4 = always*, forming two subscales labelled brooding and reflective pondering. Reflective pondering is a more adaptive way of repetitive thought processing (at least in long-term), where analyzing one's own emotions and thoughts may facilitate problem solving, while brooding can be characterized as the passive, self-criticizing dwelling on past stressful situations [2]. The sum of the scores for each subscale were used in the analyses, where higher scores indicated more usage of the specific response style. Both subscales of the Hungarian adaptation demonstrated good internal consistency in a previous study (Cronbach αs: .71 and .73, respectively) [71], as well as in the current sample (Cronbach αs were .71 for brooding and reflective pondering).

*The Zung Self-rating Depression Scale (ZSDS).* [72,73] was used to measure depressive symptoms. The ZSDS is a 20-item instrument (e.g. *"I have trouble sleeping at night"*) where each item is rated on a 4-point scale (*1 = a little of a time; 4 = most of the time*). The total score (ranged between 20–80) of ZSDS was calculated and used in the analysis, where higher scores indicates more depressive symptoms. Internal consistency of the scale was acceptable (Cronbach α = .67).

RTSQ described above was also used in Study 2.

**Statistical analysis.** Pearson correlation analyses were applied to test the construct validity of the RTSQ using Mplus 7.4. Coefficients between RRS, ZSDS and RTSQ were interpreted and the level of significance was set to .05. We examined the factor structure of the RTSQ the same way as we did in Study 1.

## Results of Study 2

**Descriptive statistics and construct validity.** Means, standard deviations, Cronbach's alphas and correlations between measures are presented in Table 8. As expected, the RTSQ showed significant positive correlations with the ZSDS total score (similarly to Study 1) and both subscales of the RRS, but the strength of the associations was considerably different for the two RRS subscales: the RTSQ total score (as well as its subscales) was weakly associated with reflective pondering (r = .23), but showed stronger positive correlations with brooding (r = .60). No significant gender differences were found.

We also performed a regression model (Table 9) to be able to control for gender and age, and to see whether ZSDS is significantly associated with RTSQ even after controlling for the RRS subscales.

Similarly to Study 1, the bifactor ESEM showed good fit to the data in Study 2 ($\chi^2$ = 169.632, df = 86, CFI = .96, TLI = .93, RMSEA = .06, SRMR = .03). Further details about CFA models and internal consistency can be found in S2 and S3 Tables.

**Table 8. Pearson correlations between measures along with the means, standard deviations and Cronbach's alphas (Study 2).**

| | 1. | 2. | 3. | 4. | 5. | 6. | 7. | 8. | 9. |
|---|---|---|---|---|---|---|---|---|---|
| 1.RTSQ total($\alpha$ = .90) | | | | | | | | | |
| 2. PfT($\alpha$ = .79) | .79** | | | | | | | | |
| 3. CT ($\alpha$ = .80) | .70** | .45** | | | | | | | |
| 4. RT($\alpha$ = .84) | .79** | .55** | .46** | | | | | | |
| 5. AT($\alpha$ = .75) | .72** | .43** | .33** | .47** | | | | | |
| 6. RRS total | .54** | .49** | .40** | .50** | .33** | | | | |
| 7.RRS Brooding($\alpha$ = .71) | .60** | .50** | .52** | .50** | .32** | .78* | | | |
| 8.RRS Reflective pondering($\alpha$ = .71) | .23** | .14* | .10 | .26** | .14* | .78** | .21** | | |
| . ZSDS($\alpha$ = .67) | .58** | .63** | .40** | .45** | .28** | .41** | .54** | .09 | |
| Total sample, M (SD) | 75.41 (19.19) | 11.12 (4.87) | 11.47 (4.74) | 20.59 (4.93) | 12.90 (4.11) | 23.59 (4.93) | 10.73 (3.17) | 12.87 (3.16) | 39.28 (7.70) |
| Males, M (SD) | 72.88 (20.24) | 10.56 (4.89) | 11.79 (5.31) | 19.63 (5.02) | 12.06 (4.02) | 23.23 (5.33) | 10.21 (3.16) | 13.02 (3.48) | 37.78 (6.32) |
| Females, M (SD) | 75.90 (18.99) | 11.23 (4.88) | 11.41 (4.63) | 20.78 (4.90) | 13.06 (4.12) | 23.66 (4.85) | 10.83 (3.17) | 12.84 (3.10) | 39.57 (7.92) |
| t-statistics (p) | 1.03 (.31) | 0.90 (.37) | 0.53 (.60) | 1.54 (.13) | 1.61 (.11) | 0.58 (.56) | 1.29 (.20) | 0.38 (.70) | 1.51 (.13) |

Total Sample: N = 320; Males: N = 52 (16%); Females: N = 268 (84%); RTSQ total, Ruminative Thought Style Questionnaire total score; RTSQ subscales: RT, Repetitive thoughts; CT, Counterfactual thinking; PfT, Problem-focused thoughts; AT, Anticipatory thoughts; RRS, Ruminative Response Scale; ZSDS, Zung Self-rating Depression Scale; M, mean; SD, standard deviation.

*p < .05

** p < .001.

## Discussion of Study 2

In Study 2, we tested the construct validity of the RTSQ compared to the RRS, one of the most extensively used rumination measures [74] and we also measured its unique relation to depressive symptoms (as assessed by the ZSDS). The analyses revealed that the RTSQ was more strongly associated with the brooding subscale than the reflective pondering subscale of the

**Table 9. Association between RTSQ total scores, trait rumination (measured by RRS) and depressed mood (ZSDS scores) after controlling for gender and age.**

| | B | SE | $\beta$ | $R^2$ | $\Delta R^2$ |
|---|---|---|---|---|---|
| Step 1 | | | | .397 | |
| Constant | 50.700 | 9.469 | | | |
| gender | .621 | 2.369 | .012 | | |
| Age | -.994 | .299 | -.153** | | |
| RRS Brooding | 3.369 | .281 | .559*** | | |
| RRS Reflection | .803 | .291 | .128** | | |
| Step 2 | | | | .479 | .081*** |
| Constant | 23.115 | 9.710 | | | |
| gender | .172 | 2.209 | .003 | | |
| age | -.717 | .282 | -.111* | | |
| RRS Brooding | 2.301 | .305 | .381*** | | |
| RRS Reflection | .797 | .272 | .127** | | |
| ZSDS total | .854 | .126 | .342*** | | |

RTSQ total, Ruminative Thought Style Questionnaire total score; RRS, Ruminative Response Scale, ZSDS, Zung Self-rating Depression Scale.

*p < .05

** p < .01

***p < .001.

RRS, thus, it appears that the RTSQ captures the maladaptive aspect of rumination more distinctively than the reflective pondering component. Our results also demonstrated significant positive association with depressive symptoms measured by the ZSDS, reaffirming its validity due to the well-known association between rumination and depressive symptoms [6]. Our results and previous empirical evidence [27] on the association of the RTSQ factors with the RRS subscales could also suggest that ruminative thoughts are associated, but not redundant with the response style assessed by RRS. Furthermore, we managed to provide further support for the findings of Study 1 regarding the factor structure of the RTSQ, as the proposed bifactor ESEM demonstrated good model fit on an independent sample.

## General discussion

The goal of our study was to explore the psychometric properties of the RTSQ. We wished to see whether rumination as an underlying construct emerged behind the different items, in other words, to see whether the RTSQ total score is a valid measure of rumination. As rumination is a transdiagnostic risk factor to psychopathology [75] that should be a target of interventions [76], it is crucial to define reliable rumination measures for assessment and treatment purposes—in case of the RTSQ, to disentangle whether the global score or the subscales are more advised to use for such purposes. This is especially important in case of a university student sample, as that age range is considered a sensitive period for the emergence of psychological problems such as mood disorders [77], for which rumination is considered a substantial risk factor, primarily among women [6], highlighting the relevance of examining gender invariance of rumination measures such as the RTSQ. Moreover, rumination has been the target of specific therapeutic interventions [78], hence a reliable rumination scale could help to accurately measure post-therapeutic change in ruminative tendencies. Prior research about the factor structure of the RTSQ was indefinite, hence we examined several previously proposed models: the unidimensional solution suggested by the authors of the RTSQ [7], the four-factor structure introduced by Tanner et al. [24], and the bifactor model presented by Mihić et al. [27]. In addition, we proposed a fourth model, a bifactor ESEM solution containing 19 items on the general factor, and 14 items on the subscales, as suggested by Mihić et al. [27].

Our results supported the bifactor ESEM solution, were most of the variance is explained by the general rumination factor. This indicates that the original aim of the authors of the RTSQ was attained, i.e. to construct a scale that assesses rumination globally [7]. To conclude, our results align with the findings of Mihić et al. [27], i.e. that the bifactor solution is the most adequate model, where the total score of the RTSQ can be used reliably, and the application of the subscales is ambiguous. We managed to provide further support to this finding on a smaller independent sample in Study 2. The differences in factor loadings may be attributed to cultural or idiomatic differences, as well as to certain sample characteristics, thus we did not find it justifiable to rule more items out based on the results of our study. Furthermore, we did not wish to strictly follow the subscales recommended by Tanner et al. [24], as they conducted their research on an adolescent sample, thus their results may not entirely apply for adults. It appears that more studies are needed to clarify the applicability of certain ambiguous items.

Our study indicated that the RTSQ is a reliable measure across genders, which is important due to the well-documented gender differences in rumination, namely that women generally report more rumination than men [e.g. 14,64]. This variation has been suggested to account for the gender difference, at least partially, in depression, i.e. that women are twice as likely to suffer from major depressive disorder during their lifespan than men [13,79]. Thus, investigating whether men and women tend to interpret the items of self-report rumination measures

equivalently is crucial for the practical implication of their results. Our results support that the gender difference in the RTSQ total score is not attributable to response bias.

In terms of internal consistency, the RTSQ total score seems to be a valid measure of ruminative thought style. Besides Cronbach alpha, the omega coefficient also supported the internal consistency of the scale. Since the omega hierarchical values were low compared to the omega total values, we could assume that most of the subscale score variances could be attributed to the general factor, and not to the group factors.

The authors of the RTSQ were aiming to design a scale that assesses rumination globally. The correlational analyses in Study 1 revealed that the RTSQ is strongly associated with general symptom severity, implying that the goal of Brinker and Dozois [7] was successfully attained. The correlations in Study 2 demonstrated that the RTSQ is congruent with one of the most extensively used rumination measures, the RRS [74]. Moreover, it revealed that the thought style captured by the RTSQ is rather maladaptive, as it was more strongly associated with the brooding subscale of the RRS than with reflective pondering. Brooding, the maladaptive facet of rumination, defined as a tendency to passively dwell on negative emotions (i.e. What am I doing to deserve this?) was more strongly associated with concurrent distress than reflective pondering (the latter defined as a purposeful self-reflective response of understanding and solving the problem) [2]. In addition, brooding also related to depression scores prospectively [2,80], while reflective pondering (or reflection) did not. Studies that tested the unique contributions of brooding and reflective pondering to different internalizing or externalizing symptoms and disorders found that brooding is the most maladaptive (even pathological) form of depressive rumination [15,81], while reflective pondering could serve as a protective factor against the detrimental effects of these unconstructive, often self-deprecating thoughts [17,82]. However, recalling negative events and affects, even in this adaptive way, could temporally elevate the level of negative emotions, which could explain why reflective pondering is significantly associated with concurrent distress in cross-sectional studies [71,83]. Taken together, our results support the construct and convergent validity of the RTSQ, indicating that it is a valid measure of ruminative thinking style that is rather negatively valenced, but does not solely focus on depressive mood and symptoms. From a theoretical point of view, it is important to mention that Tanner and colleagues' solution on the four facets of the 15-item version could be considered as an attempt to identify key dimensions of ruminative thinking. Watkins and Roberts [15] in their recent review, for instance, claim that besides the frequency of ruminative thoughts, other relevant dimensions of ruminative thinking should be targeted. Based on the habit-goal theory, rumination can easily become a mental habit if this maladaptive thinking repeatedly occurs in the same context (including mood, social event or physical location) [84]. Watkins and Roberts [15] mention that automaticity, involuntariness, and goal insensitivity are of great relevance. Whereas certain items are not always found to belong to the same subscale, the *Repetitive thoughts* (items 1–4) subscale of the RTSQ has been consistently identified by numerous psychometric studies [e.g. 24,26,27], as well as in our study, suggesting that the RTSQ captures well the repetitive nature of ruminative thinking. Many papers emphasize the repetitive nature of rumination, making it an example of repetitive negative thinking [85].

A strength of our research is that we conducted two consecutive studies with converging results on two homogenous samples of university students, as opposed to the more heterogeneous samples observed in previous studies. However, it is a limitation that most of the participants were female, especially in Study 2, where the sample size was also much smaller. Although our sample comprised of university students that may reduce generalizability, we consider it important to examine rumination among young adults, as rumination and depressive symptoms are commonly observed in this population [86,87]. Another limitation is that

the models we wished to replicate were tested on different translations of the RTSQ (e.g. English and Serbian), whereas we tested the factor structure of the Hungarian translation. Although beyond the scope of our work, it would be crucial for future studies to investigate whether diverging results reflect inconsistency in the measurement of rumination as a construct per se, or rather reflect idiomatic differences. Furthermore, we did not examine the discriminant validity of the RTSQ. Although rumination is associated with a wide array of psychological (and somatic) problems [e.g. 15,71], Agreeableness, a Big Five personality trait defined as accommodating, amiable, friendly, and trustworthy [88] appears to be an unrelated construct [89] that could be used for such purposes. However, this was beyond the scope of our current paper.

To sum up, our results demonstrate that the Hungarian adaptation of the RTSQ reliably measures rumination across gender, and it can be considered a valid measure to assess ruminative thinking in general with its total score, meanwhile the use of its subscales is ambiguous. Moreover, the global RTSQ score appears to primarily measure the maladaptive aspect of rumination, hence, it can be associated with psychopathology in general.

## Supporting information

**S1 Table. Standardized regression weights between RTSQ total scores and covariates.** Total Sample: N = 1123; Males: N = 505 (45%); Females: N = 618 (55%). RTSQ, Ruminative Thought Style Questionnaire total score; CES-D, The Center for Epidemiologic Studies Depression Scale; BSI_GSI, Brief Symptom Inventory General Symptom Index; PfT, Problem-focused thoughts factor of the Ruminative Thought Style Questionnaire; CT, Counterfactual thinking factor of the Ruminative Thought Style Questionnaire; RT, Repetitive thoughts factor of the Ruminative Thought Style Questionnaire; AT, Anticipatory thoughts factor of the Ruminative Thought Style Questionnaire. $^*$p < .05.; $^{**}$p < .01.; $^{***}$p < .001.
(DOCX)

**S2 Table. Factor analyses of four measurement models of the Ruminative Thought Style Questionnaire in Study 2.** Model 1 = One factor CFA; Model 2 = Second-order four factor CFA; Model 3 = bifactor CFA; Model 4 = bifactor ESEM; AIC, Akaike Information Criteria; $\chi 2$, chi-square test statistic; df, degree of freedom; CFI, Comparative Fit Index; TLI, Tucker-Lewis Index; RMSEA, Root Mean Squared Error of Approximation; CI, confidence interval; SRMR, Standardized Root Mean Square Residual.
(DOCX)

**S3 Table. Alpha and Omega reliability for the bifactor ESEM (Model 4) in Study 2.** RT, Repetitive thoughts factor of the Ruminative Thought Style Questionnaire; CT, Counterfactual thinking factor of the Ruminative Thought Style Questionnaire; PfT, Problem-focused thoughts factor of the Ruminative Thought Style Questionnaire; AT, Anticipatory thoughts factor of the Ruminative Thought Style Questionnaire.
(DOCX)

## Acknowledgments

We would like to thank everyone who helped our work by participating in this study.

## Author Contributions

**Conceptualization:** Gyöngyi Kökönyei.

**Data curation:** Lilla Nóra Kovács, Natália Kocsel.

**Formal analysis:** Lilla Nóra Kovács, Natália Kocsel.

**Investigation:** Natália Kocsel, Attila Galambos, Anna Magi.

**Methodology:** Lilla Nóra Kovács, Natália Kocsel, Gyöngyi Kökönyei.

**Project administration:** Gyöngyi Kökönyei.

**Supervision:** Zsolt Demetrovics, Gyöngyi Kökönyei.

**Writing – original draft:** Lilla Nóra Kovács, Natália Kocsel.

**Writing – review & editing:** Attila Galambos, Anna Magi, Zsolt Demetrovics, Gyöngyi Kökönyei.

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
