## [Decision Letter · Decision Letter 0]

18 May 2021

PONE-D-21-01374

Validating the Bifactor Structure of the Ruminative Thought Style Questionnaire - a Psychometric Study

PLOS ONE

Dear Dr. Kökönyei,

Thank you for submitting your manuscript to PLOS ONE. After careful consideration, we feel that it has merit but does not fully meet PLOS ONE’s publication criteria as it currently stands. Therefore, we invite you to submit a revised version of the manuscript that addresses the points raised during the review process.

We look forward to receiving your revised manuscript.

Kind regards,

Frantisek Sudzina

Academic Editor

PLOS ONE

Journal Requirements:

2) Peer review at PLOS ONE is not double-blinded (https://journals.plos.org/plosone/s/editorial-and-peer-review-process). For this reason, authors should include in the revised manuscript all the information removed for blind review, including URLs.

Reviewers' comments:

Reviewer's Responses to Questions

**Comments to the Author**

1. Is the manuscript technically sound, and do the data support the conclusions?

Reviewer #1: No

Reviewer #2: Partly

2. Has the statistical analysis been performed appropriately and rigorously? 

Reviewer #1: No

Reviewer #2: No

3. Have the authors made all data underlying the findings in their manuscript fully available?

Reviewer #1: Yes

Reviewer #2: Yes

4. Is the manuscript presented in an intelligible fashion and written in standard English?

Reviewer #1: Yes

Reviewer #2: No

5. Review Comments to the Author

Reviewer #1: The present study attempts to investigate the factor structure of the RTSQ using ESEM bifactor in two large samples, as well as gender invariance and criterion-related validity. There are a number of technical/methodological problems with this article. Additionally, the overall scientific contribution is not clear. I would suggest making the paper a brief report.

To begin, the introduction/setup of the paper was far too long. The authors should remember that factor analysis does not ‘uncover reality’. In this vein, detailed descriptions of who found which factor structure are likely not needed. Moreover, what is not clear is the ultimate contribution of this study. What is the clinical relevance of finding that the RTSQ has this structure or that? How does one versus the other finding in terms of structure impact what you do in the clinic?

Notably, page 12, line 179 – predicting depression from the RTSQ factors is criterion-related validity, not convergent validity (that would be another measure of rumination).

Methods:

Please document – at least briefly – the validation procedure for Hungarian measures in terms of translation/back-translation and cross-cultural construct validity previously found, if any.

It seems a little odd to use fit indices to arbitrate between models, especially when using a bifactor and ESEM. The bifactor is already a very general model and has a tendency to overfit – there’s quite a bit of discussion in the literature about it. ESEM compounds this problem further; in other words, there’s basically no way for the ESEM bifactor not to come out as the best-fitting model. The authors should at least discuss some of the auxiliary ways of arbitrating between models – content of general/specific factors, whether a factor is ‘collapsing in’ on itself, whether it’s driven by a single factor loading (see the AT factor), etc in this context.

Next, the authors could have reasonably stopped at Model 3 – it’s more interpretable and fits fine (although I’d have to look at the specific factor loadings to be sure). Additionally, why use 90% rather than 95% CI? Finally, the cross-loadings don’t seem to be reported (Table 3).

Notably, please go through the measure carefully and clean out any criterion contamination with depression items or criteria. Some of the content still appears overlapping. Additionally, in Table 3, items 18 and 20 seem to be the same.

There appear to be issues with the gender invariance models. First, the authors don’t include the AIC across models. Second, it appears that the likelihood ratio tests across models are significant, which would indicate a worsening fit across more constrained models. As such, I don’t think the authors can argue for gender invariance. Finally, why not examine predictive invariance (predicting depression) across gender?

Why not examine whether the specific factors predict a criterion variable above and beyond the general factor within the model, rather than just using sum scores?

Study 2: why did the authors not replicate the same models as in Study 1? Also, why was discriminant validity (e.g., some externalizing metric) not included?

Age should have been controlled for, given the range.

Reviewer #2: This manuscript investigates psychometric properties of the Ruminative Thought Style Questionnaire (RTSQ) among university students. Authors found that (1) the factor structure of the RTSQ was best described with a 19-item bifactor Exploratory Structural Equation Modelling (ESEM), and (2) the RTSQ was congruent with the RRS. This study appears to have a good methodology; however, it is unclear what the rumination would be.

Introduction

1.The most important problem was the lack of information about the definition of "Rumination." What is the gold standard for this word?

2.Although authors state that “Ruminative response, accompanied by other psychosocial factors, appears to aggravate depressed mood from early adolescence in case of women (ref.10)”, are there any reports that have shown clinical significance using the RTSQ (score) in university students? The significance of the RTSQ total score in the target population should be justified in the Introduction section, since the authors state that they want to see if the RTSQ total score is a valid measure of rumination.

3.The authors raise the ambiguity of the concept of rumination, and issues of differences in structural validity between studies as a problem, but there is not enough information on a problem for content validity of the RTSQ.

Methods & Results;

4.Sample collection: How were the participants collected for this study？ At least, eligibility criteria need to be shown. (Study1 & Study2)

5.Convergent validity needs to be assessed in separate analyses for men and women, respectively. (Study1)

6.Why did the authors examine the correlation between the CES-D and RTSQ in Study 1, while the correlation between the ZSDS and RTSQ in Study 2? Not only the ZSDS but also the CES-D scale are widely used to evaluate depressive symptom severity, with higher scores reflecting increased symptom severity. Please elaborate on why the depression scale was changed between studies 1 and 2. (Study1 & Study2)

7.Why did authors examine the relationship between the RTSQ scores and each subscale score, but not the total RPS score? (Study2)

Discussion;

8.In the method section, the word "Reliabilty" is used when explaining Cronbachα, but the word "Internal consistency" is used in the discussion section. The different ways of expressing the same meaning can be confusing. According to the COSMIN, the word "Reliability" can have several meaning, which may cause misinterpretation.

6. PLOS authors have the option to publish the peer review history of their article (what does this mean?). If published, this will include your full peer review and any attached files.

Reviewer #1: No

Reviewer #2: No

---

## [Author Response · Author response to Decision Letter 0]

5 Jul 2021

Dear Dr. Frantisek Sudzina,

We thank you that you are willing to consider a revision. We would like to take the opportunity and resubmit the revised version of our manuscript, entitled “Validating the Bifactor Structure of the Ruminative Thought Style Questionnaire - a Psychometric Study”.

Please, find our detailed answers to the reviewers’ comments in the uploaded “Response to reviewers” file, alongside with the revised manuscript. 

We would like to thank our reviewers for their time reviewing our manuscript that helped us improve our work substantially. We hope we have managed to address each of their concerns.

Sincerely,

the authors

---

## [Editor Report · Decision Letter 1]

8 Jul 2021

Validating the Bifactor Structure of the Ruminative Thought Style Questionnaire - a Psychometric Study

PONE-D-21-01374R1

Dear Dr. Kökönyei,

We’re pleased to inform you that your manuscript has been judged scientifically suitable for publication and will be formally accepted for publication once it meets all outstanding technical requirements.

Kind regards,

Frantisek Sudzina

Academic Editor

PLOS ONE
---

## [Editor Report · Acceptance letter]

16 Jul 2021

PONE-D-21-01374R1 

Validating the Bifactor Structure of the Ruminative Thought Style Questionnaire - a Psychometric Study 

Dear Dr. Kökönyei:

I'm pleased to inform you that your manuscript has been deemed suitable for publication in PLOS ONE. Congratulations! Your manuscript is now with our production department. 

Kind regards, 

on behalf of

Dr. Frantisek Sudzina 

Academic Editor

PLOS ONE